# GAUSSIAN-DET: LEARNING CLOSED-SURFACE GAUSSIANS FOR 3D OBJECT DETECTION

**Hongru Yan**[*], **Yu Zheng**[*], **Yueqi Duan**[†]
Tsinghua University
`yanhr21@mails.tsinghua.edu.cn,`
`yu-zheng@tsinghua.edu.cn,`
`duanyueqi@tsinghua.edu.cn`

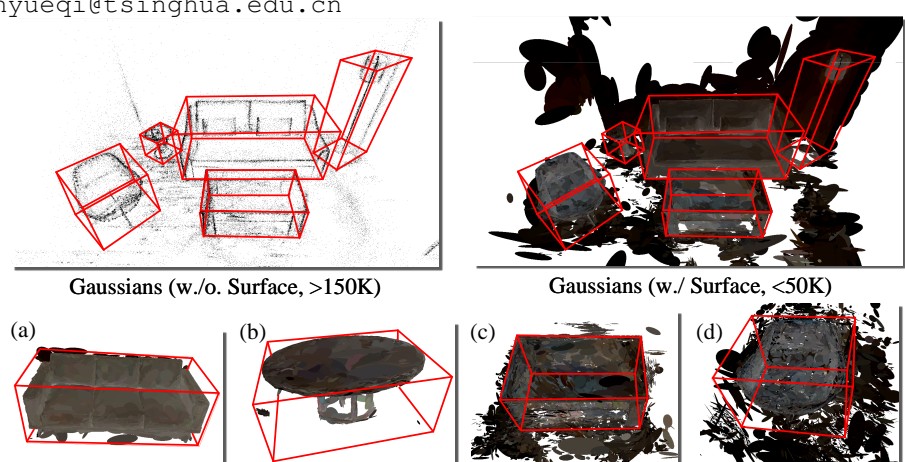

Figure 1: We leverage Gaussian Splatting as surface representation for multi-view based 3D object detection. **Top Left:** Gaussians with color and coordinate. **Top Right:** Gaussians with color, coordinate and surface information, but in a much fewer amount. **Bottom (a-b):** Object composed of Gaussians exhibits a textured surface as opposed to the top left illustration. **Bottom (c-d):** However, Gaussian splatting inherently introduces numerous outliers around the object.

## ABSTRACT

Skins wrapping around our bodies, leathers covering over the sofa, sheet metal coating the car – it suggests that objects are enclosed by a series of continuous surfaces, which provides us with informative geometry prior for objectness deduction. In this paper, we propose Gaussian-Det which leverages Gaussian Splatting as surface representation for multi-view based 3D object detection. Unlike existing monocular or NeRF-based methods which depict the objects via discrete positional data, Gaussian-Det models the objects in a continuous manner by formulating the input Gaussians as feature descriptors on a mass of partial surfaces. Furthermore, to address the numerous outliers inherently introduced by Gaussian splatting, we accordingly devise a Closure Inferring Module (CIM) for the comprehensive surface-based objectness deduction. CIM firstly estimates the probabilistic feature residuals for partial surfaces given the underdetermined nature of Gaussian Splatting, which are then coalesced into a holistic representation on the overall surface closure of the object proposal. In this way, the surface information Gaussian-Det exploits serves as the prior on the quality and reliability of objectness and the information basis of proposal refinement. Experiments on both synthetic and real-world datasets demonstrate that Gaussian-Det outperforms various existing approaches, in terms of both average precision and recall.

## 1 INTRODUCTION

3D object detection in indoor scenes (Dai et al., 2017; Hu et al., 2023; Rukhovich et al., 2022b) has been a popular topic in industry and academia, fueling various applications such as robotic navigation and augmented reality. While many works (Qi et al., 2019; Liu et al., 2021; Cheng et al., 2021; Rukhovich et al., 2022a; Ran et al., 2022) have demonstrated the effectiveness on point cloud

---

[*]Equal Contributions.    [†] Corresponding Author.

Figure 2: Illustration of the surface closure prior in Gaussian-Det. For 3D Gaussians **within a bounding box**, those forming relatively closed surfaces (lower $|\Phi|$) indicate an accurate detection, and vice versa. Note that 3D Gaussians outside the bounding box are not shown for clarity.

data captured by expensive sensors such as lidars and depth cameras, RGB camera systems offer a cost-effective alternative. However, the absence of depth information poses major challenges for inferring the three-dimensional geometries of the target objects from two-dimensional visual clues.

To overcome the aforementioned ill-posedness in image-based 3D object detection, pure multi-view based approaches (Rukhovich et al., 2022b; Tu et al., 2023) projects the features on the image plane into a 3D volume representation. However, such projective geometry is still dominated by 2D features which may lead to ambiguous detection results. On the other hand, recent advances in Neural Radiance Field (NeRF) (Mildenhall et al., 2020) enjoy multi-view consensus and consistency via learning a plenoptic mapping function. Leveraging NeRF as the 3D scene representation, NeRF-RPN (Hu et al., 2023) consumes volumetric information extracted from NeRF and directly outputs 3D bounding boxes through a general region proposal network (RPN). However, due to the implicit and continuous representation of NeRF, these methods are computationally expensive to optimize. More importantly, similar to monocular-based methods (Rukhovich et al., 2022b; Tu et al., 2023), they depend on uniform and discrete sampling in 3D space which may lead to sub-optimal performance by failing to capture the true objectness. In contrast, as indicated by the bottom (a-b) part of Figure 1, real-world 3D objects are usually enclosed by a series of continuous surfaces. Such surface information provides informative visual clues for objectness deduction (Ren and Sudderth, 2018).

In this paper, we propose Gaussian-Det which exploits continuous surface representation for multi-view based 3D object detection. Gaussian-Det leverages the recent Gaussian Splatting (Kerbl et al., 2023) and formulates the input Gaussians into a mass of descriptors on partial surfaces. As shown by top of Figure 1, even with less than a third of the number, Gaussians utilizing surface information better demonstrate the texture information, smoothness and continuity. While it is feasible to directly employ point cloud based detectors, they can be distracted by numerous outliers inherently introduced by Gaussian Splatting process shown in Bottom (c-d) part of Figure 1. Moreover, the point-based detectors regard the input Gaussians as infinitesimal positional representations and discard the explicit depiction on continuous surface, which may lead to sub-optimal detection proposals. To address this, we design a Closure Inferring Module (CIM) for the comprehensive surface-based objectness deduction in Gaussian-Det. As displayed in Figure 2, CIM is built upon the assumptive prior that, the partial surfaces in an accurately detected bounding box can form a relatively closed surface measured by a lower flux value. Specifically, taking the underdetermined nature of Gaussian Splatting into account, CIM firstly conducts the partial surface inference via estimating a variational term as a probabilistic feature residual to the original representation of partial surfaces. CIM then coalesces them into a holistic representation and uses the flux value measures the closure of the overall proposal, serving as the prior on the quality and reliability of objectness prediction and the information basis of proposal refinement. To validate the effectiveness of the proposed Gaussian-Det, we have conducted experiments on both synthetic 3D-FRONT (Fu et al., 2021) and real-world ScanNet (Dai et al., 2017) datasets. The experimental results demonstrate that Gaussian-Det outperforms various existing approaches in terms of both average precision and recall.

## 2 RELATED WORK

**Neural Scene Representation:** In recent years, neural scene representation have presented astonishing performances on visual computing and geometry learning. They usually take a coordinate as input and output corresponding scene properties, such as signed distance (Park et al., 2019; Chabra et al., 2020), occupancy (Mescheder et al., 2019), color and density. Notably, in the scenario of multi-view geometry reconstruction, the Neural Radiance Field (NeRF) (Mildenhall et al., 2020)

learns a plenoptic mapping from the querying coordinate and viewing direction into the point-wise radiation and volume density. The NeRF-based scene modeling has inspired numerous extended works spanning from low-level geometry reconstruction (Barron et al., 2021; Fridovich-Keil et al., 2022; Wang et al., 2021) to high-level visual understanding (Zhi et al., 2021; Jeong et al., 2022; Hu et al., 2023; Xu et al., 2023; Liu et al., 2023; Tian et al., 2024; Irshad et al., 2024). Since NeRF requires a massive amount of queries during both training and inference phases, some improved representations are introduced as proxies such as hash grids (Müller et al., 2022), tri-planes (Chen et al., 2022) and points (Xu et al., 2022a). Among these advancements, the 3D Gaussian Splatting (3D-GS) (Kerbl et al., 2023) stands out for its balance between quality and efficiency. 3D-GS employs a set of anisotropic Gaussians initialized by Structure from Motion (SfM), whose amount and properties are iteratively updated during training. Apart from the positional information as in point cloud based or NeRF-based representation, a 3D Gaussian naturally depicts a continuous surface through its mean, covariance parameters. Based on 3D-GS, several works have explored on novel view synthesis Charatan et al. (2024); Chen et al. (2024b); Zhang et al. (2024), 3D content generation (Tang et al., 2024), surface reconstruction (Guédon and Lepetit, 2024), semantic segmentation (Shi et al., 2024) and editing (Chen et al., 2024a).

**Indoor 3D Object Detection:** Indoor 3D object detection has drawn extensive attention owing to its broad applications in robotic vision, augmented reality, etc. The techniques can be categorized according to the input modalities, such as point cloud, gridded voxels and 2D image(s). For example, (Song and Xiao, 2016) uses a volumetric CNN to create 3D Region Proposal Network (RPN) on a voxelized 3D scene. VoteNet (Qi et al., 2019) directly operates on point cloud by using a PointNet++ (Qi et al., 2017b) backbone and center voting strategy. The follow-up point-based approaches include the VoteNet-style (Cheng et al., 2021; Liu et al., 2021; Zheng et al., 2024) and the sparse convolution based methods (Rukhovich et al., 2022a; Wang et al., 2022). However, the reliance on expensive lidars or depth sensors limits their applicability. In contrast, the RGB camera systems offer a cost-effective alternative. To overcome the inherent ill-posedness in image-based 3D object detection, ImVoxelNet (Rukhovich et al., 2022b) lifts the 2D features into a 3D volume representation followed by a FCOS (Tian et al., 2019)-style detection head. ImGeoNet (Tu et al., 2023) introduces an image-induced geometry-aware voxel representation for better preservation of geometry. However, the projective geometry extracted by (Rukhovich et al., 2022b; Tu et al., 2023) still originates from 2D features and may thus result in ambiguous detections. On the other hand, several advances (Hu et al., 2023; Xu et al., 2023; Irshad et al., 2024) seek to leverage the multi-view consensus and consistency of NeRF representation (Mildenhall et al., 2020) that embeds the geometry into the voxel grids. NeRF-RPN consumes the volumetric features sampled from NeRF and then directly regresses the bounding boxes via a generic RPN. NeRF-Det learns a two-branch framework that simultaneously conducts geometry reconstruction and bounding box regression. These two approaches rely on uniform and discrete sampling which may fail to capture the true objectness in 3D scenes. In contrast, our proposed Gaussian-Det strives for representing the targets in a continuous manner by leveraging the descriptors on partial surfaces and coalescing them into a holistic measurement which serves as a comprehensive prior on the reliability of the deducted objectness.

## 3 APPROACH

In this section, we elaborated the proposed Gaussian-Det by dividing the framework into the construction of surface-based Gaussian representation, object proposal initialization, partial surface feature inference and holistic surface closure coalescence. The overall framework is illustrated in Figure 3.

### 3.1 PRELIMINARY: 3D GAUSSIAN SPLATTING

The input scene of our proposed Gaussian-Det is represented by 3D Gaussian Splatting (3D-GS) (Kerbl et al., 2023) that are reconstructed from posed images, comprising a set of 3D Gaussian primitives $\mathbb{G} = \{\mathbf{g}_p = (\mu_p, \Sigma_p, \alpha_p, \mathbf{c}_p)\}_p^P$. Each 3D Gaussian $\mathbf{g}_p$ is parameterized by a mean vector $\mu_p \in \mathbb{R}^3$ indicating its center, an anisotropic covariance matrix $\Sigma_p \in \mathbb{R}^{3\times3}$, the opacity value $\alpha_p \in \mathbb{R}$ and color $\mathbf{c}_p \in \mathbb{R}^3$. $\Sigma_p$ is a positive semidefinite matrix and can be factorized as $\Sigma_p = \mathbf{R}_p \mathbf{S}_p \mathbf{S}_p^T \mathbf{R}_p^T$. $\mathbf{R}_p \in \mathbb{R}^{3\times3}$ and $\mathbf{S}_p \in \mathbb{R}^{3\times3}$ refer to a diagonal size matrix $diag(s_p^1, s_p^2, s_p^3)$ and a rotation matrix converted from a unit quaternion respectively. Subsequently, the 3D Gaussians are projected onto the image plane using known camera matrices. $\alpha$-blending is executed to aggregate the overlapping

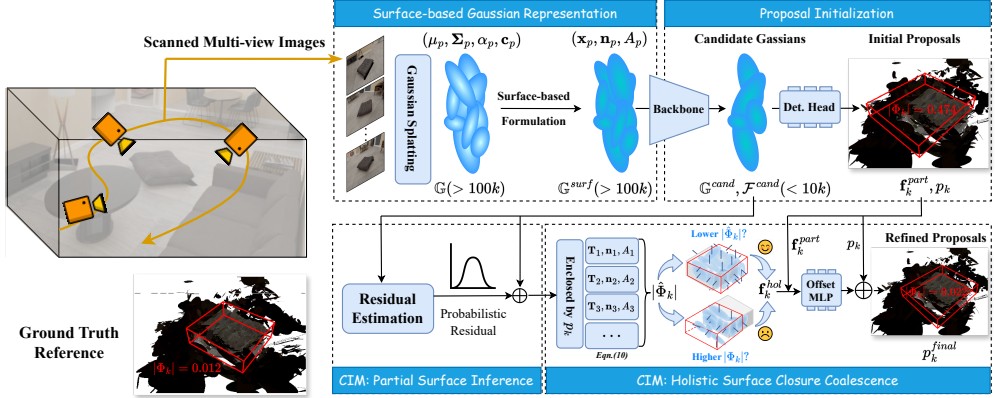

Figure 3: From the original Gaussians $\mathbb{G}$, we formulate surface-based Gaussian representation $\mathbb{G}^{surf}$, which is firstly used to predict the initial object proposal with opener surfaces. Then the closure inferring module, which contains partial surface feature inference and holistic surface closure coalescence. CIM learns a informative prior on the quality of the predicted objectness, thus controlling the support (☺) or suppression (☹) of the holistic object representation. The partial and holistic representations are combined to estimate the refined detection result, whose degree of surface closure is similar to that of ground truth measured by the absolute flux value $|\Phi_k|$.

Gaussians on each pixel and compute the final color:

$$C = \sum_{i=1}^{n} \mathbf{c}_i \alpha_i \prod_{j=1}^{i-1} (1 - \alpha_j) \tag{1}$$

where $n$ is the number of overlapping Gaussians. The attributes of $\mathbf{g}_p$ are optimized through backward propagation of the gradient flow. Particularly, the density (i.e., $P$) of 3D Gaussians is also updated during training via cloning and splitting for the sake of rendering quality.

## 3.2 SURFACE-BASED GAUSSIAN REPRESENTATION AND PROPOSAL INITIALIZATION

Compared with NeRF-based (Mildenhall et al., 2020) representation, 3D Gaussian Splatting enjoys the real-time rendering speed owing to a specially designed rasterizer. More importantly, a 3D Gaussian $\mathbf{g}_p$ approximates the shape of a bell curve with depiction on a continuous surface (especially for a flat Gaussian (Guédon and Lepetit, 2024)), as apposed to the discrete representation of infinitesimal positional data such as point cloud (Rukhovich et al., 2022a) and uniformly sampled volumes (Rukhovich et al., 2022b; Xu et al., 2023; Hu et al., 2023). To this end, we concentrate the Gaussian representation onto surface-related information by formulating:

$$\mathbb{G}^{surf} := \{\mathbf{g}_p^{surf} = (\mathbf{x}_p, \mathbf{n}_p, A_p)\}_p^P \in \mathbb{R}^{P \times 7} \tag{2}$$

where $\mathbf{x}_p = \mu_p$, $\mathbf{n}_p$ and $A_p$ specify the information about location, perpendicular orientation and area respectively. The normal vector $\mathbf{n}_p$ is determined by solving the eigenvalue problem $\mathbf{\Sigma}_p \mathbf{n}_p = \lambda \mathbf{n}_p$. Specifically, $\mathbf{n}_p$ is the eigenvector corresponding to the smallest eigenvalue. The in-ward or out-ward orientation is determined by setting the orientation of $\mathbf{n}_p$ to align with the orientation from the center of the object proposal to the Gaussian mean. Please refer to the Appendix pages for the derivative process. $A_p$ is defined as the largest cross-section area of 3D Gaussian Ellipsoid:

$$A_p = \pi \frac{s_p^1 \cdot s_p^2 \cdot s_p^3}{\min(s_p^1, s_p^2, s_p^3)} \tag{3}$$

The reformulated Gaussian representation $\mathbb{G}^{surf}$ initializes the depiction on multiple partial surfaces. Given the huge amount of Gaussians in $\mathbb{G}^{surf}$, we use an off-the-shelf backbone $\mathcal{B}(\cdot)$ for simultaneously trimming them into a candidate subset and extracting their corresponding features:

$$\mathbb{G}^{cand}, \mathcal{F}^{cand} = \mathcal{B}(\mathbb{G}^{surf}) \tag{4}$$

where $\mathbb{G}^{cand} = \{\mathbf{g}_1, \mathbf{g}_2, \ldots, \mathbf{g}_M\} \in \mathbb{R}^{M \times 7}$ is a subset of $\mathbb{G}^{surf}$ and $\mathcal{F}^{cand} = \{\mathbf{f_1}, \mathbf{f_2}, \ldots, \mathbf{f_M}\} \in \mathbb{R}^{M \times C}$ denotes the corresponding feature in the latent space. From $\mathbb{G}^{cand}$, we then accordingly

initialize the partial-aware object proposals $\mathbf{P}^{part} = \{p_1, p_2, \ldots, p_K\}$ by employing a VoteNet (Qi et al., 2019)-style head (Cheng et al., 2021) which groups $\mathbb{G}^{cand}$ in Euclidean space and predicts the bounding boxes with a MLP-based architecture (Qi et al., 2017a). Meanwhile, the partial-aware features of each proposal is extracted:

$$\mathbf{f}_k^{part} = \mathcal{M}^{part}(\mathcal{F}^{cand}, p_k) \tag{5}$$

where $\mathcal{M}^{part}$ is a querying-and-pooling function (Cheng et al., 2021) that operates around the location of each proposal.

As mentioned above, it is feasible to directly employ point cloud based detectors (Qi et al., 2019; Rukhovich et al., 2022a; Liu et al., 2021) and regard $\mathbb{G}^{surf}$ as a mass of point cloud concatenated with the corresponding feature. However, as shown in Figure 1, there still severely exists Gaussian outliers around the object surfaces that may distract the point-based detectors. We also demonstrate in Experiment section that this leads to incorrect objectness deduction and inferior detection performances evaluated on $\mathbf{P}^{part}$.

To address this, we devise a Closure Inference Module (CIM) in Gaussian-Det responsible for comprehensive surface-based objectness deduction in Gaussian-Det, which takes the holistic closure measurement into account to refine the object proposals.

### 3.3    CLOSURE INFERENCE MODULE FOR PROPOSAL REFINEMENT

As illustrated in the bottom part of Figure 3, the Closure Inference Module (CIM) consists of two stages: Given the underdetermined nature of 3D Gaussian Splatting, CIM firstly enhances the representation of partial surfaces via variationally estimating the probabilistic feature residuals. Subsequently, under the assumption that the higher degree of surface closure indicates a more accurate detection (and vice versa), CIM seeks to a holistic surface closure coalescence, serving as an informative prior on the quality of the predicted objectness.

**Partial Surface Inference:**  In Gaussian Splatting, inferring local surface features for spatial points is underdetermined due to the supervision from 2D signals, leading to unstructured outliers (Guédon and Lepetit, 2024) with unreliable local surface descriptors. To this end, we model such randomness in CIM by incorporating a variational term $\mathcal{F}^V$ into the candidate features $\mathcal{F}^{cand}$, establishing a probabilistic feature residual:

$$\mathcal{F}^{cand} \leftarrow \mathcal{F}^{cand} + \alpha \mathcal{F}^V \tag{6}$$

where $\alpha$ is a weight hyper-parameter (in practice we set $\alpha = 0.1$). The randomness is modeled by assuming $\mathcal{F}^V$ can be sampled from $\mathbf{p}_\theta(\mathcal{F}^V) = N(\mathcal{F}^V; \mathbf{0}, \mathbf{I})$. To fit the posterior $\log q_\phi(\mathcal{F}^V | \mathcal{F}^{cand})$ and likelihood $p_\theta(\mathcal{F}^{cand} | \mathcal{F}^V)$ parameterized by $\phi$ and $\theta$ respectively, the following evidence lower bound (ELBO) is optimized for the residual estimation(Kingma and Welling, 2014):

$$\mathcal{L}_{res} = \mathcal{L}_{\boldsymbol{\mu_g}, \sigma_g}(\mathcal{F}^{cand}) = \mathbb{E}_{q(\mathcal{F}^V | \mathcal{F}^{cand})}[\log p(\mathcal{F}^{cand} | \mathcal{F}^V)] - \mathrm{KL}(q(\mathcal{F}^V | \mathcal{F}^{cand}) || p(\mathcal{F}^V)) \tag{7}$$

By modeling the variational term, CIM explicitly represents this uncertainty, thereby better describing the distribution of local surface features.

**Holistic Surface Closure Coalescence:**   The partial surfaces are then coalesced into a holistic representation by CIM. Specifically, as illustrated in Figure 2, CIM leverages a key geometric property that the partial surfaces corresponding to an object can approximate a closed surface. As formulated by Theorem 1 (see Appendix for proof), this property can be mathematically expressed that the net flux $\phi$ of a constant vector field through a closed surface is zero.

**Theorem 1.** *Given a constant vector field* $\mathbf{T}$ *and a closed surface* $\mathbf{S}$*, the flux* $\boldsymbol{\Phi}$ *of the vector field* $\mathbf{T}$ *through the closed surface* $\mathbf{S}$*, also expressed as the surface integral of the* $\mathbf{T}$ *over* $\mathbf{S}$ *is zero:*

$$\boldsymbol{\Phi} = \oiint_S \mathbf{T} \cdot \mathbf{n} \mathrm{d}S = 0 \tag{8}$$

In CIM, we utilize this theorem by computing the flux of a pre-set constant vector field over the partial surfaces enclosed by an object proposal. The closer the total flux is to zero, the better the surfaces form a closed object, suggesting a higher-quality detection. To practically implement this concept for the $k$-th object proposal, CIM computes the estimated flux $\hat{\boldsymbol{\Phi}}_k$ numerically using quadrature:

$$\hat{\mathbf{\Phi}}_k = \sum_{\mathbf{g}_i \in \mathbb{G}_k^{cand}} \mathbf{T} \cdot \mathbf{n}_i A_i, \text{ where } \mathbb{G}_k^{cand} = \{\mathbf{g}_i \mid i \in \mathcal{I}_k\} \in \mathbb{R}^{N_k \times 7} \tag{9}$$

where $\mathcal{I}_k$ is a set containing the indices of the candidate Gaussians within the proposal $p_k$. $N_k$ is the number of candidate Gaussians within $p_k$. Without loss of generality, we set $\mathbf{T}$ as $(\frac{1}{\sqrt{3}}, \frac{1}{\sqrt{3}}, \frac{1}{\sqrt{3}})$.

With the estimated flux as the quantitative measurement of surface closure, CIM then formulate the geometrically coalesced representation as:

$$\mathbf{f}_k^{hol} = S_k \cdot \mathcal{M}^{hol}(\mathcal{F}^{cand}, p_k), \text{ where } S_k = \exp(-\gamma_k |\hat{\mathbf{\Phi}}_k|) \tag{10}$$

where $\mathcal{F}^{cand}$ is the enhanced feature with the probabilistic residual in Eqn.(6). $\gamma_k$ is a flexibility coefficient mapped from $\mathcal{F}^V$ through a MLP and sigmoid activation function. $\mathcal{M}^{hol}$ is another querying-and-pooling function sharing the same architecture with $\mathcal{M}^{part}$. $S_k$ serves as the closure weight parameter which controls the support ($|\hat{\mathbf{\Phi}}_k| \downarrow, S_k \uparrow$) or suppression ($|\hat{\mathbf{\Phi}}_k| \uparrow, S_k \downarrow$) of $\mathbf{f}_k^{hol}$.

We finally formulate the refined proposal feature which comprehensively combines the information on partial-aware surfaces and holistic closure-based measurement on objectness as:

$$\mathbf{f}_k^{final} := [\mathbf{f}_k^{part} || \mathbf{f}_k^{hol}] \tag{11}$$

where $[\cdot||\cdot]$ denotes the concatenation along the feature dimension. $\mathbf{f}_k^{final}$ is used for proposal refinement by estimating a parameter offset to the original proposals through a stacked MLP:

$$p_k^{refined} = p_k + MLP(\mathbf{f}_k^{final}) \tag{12}$$

Given the final refined object proposals $p_k^{refined}$, the overall loss function is computed as:

$$\begin{aligned} \mathcal{L}(\{p_k^{gt}\}, \{p_k\}, \{p_k^{refined}\}) = \lambda_{res}\mathcal{L}_{res} &+ \lambda_{pred}\mathcal{L}_2(\{p_k^{gt}\}, \{p_k\}) \\ &+ \lambda_{refine}\mathcal{L}_2(\{p_k^{gt}\}, \{p_k^{refined}\}) \end{aligned} \tag{13}$$

where $\{p_k^{gt}\}$ and $\mathcal{L}_2$ denote the ground truth bounding boxes and L2 norm loss function. $\lambda_{res}$, $\lambda_{pred}$ and $\lambda_{refine}$ are pre-set weight hyper-parameter.

## 4 EXPERIMENTS

We validate the proposed Gaussian-Det by conducting benchmark experiments on both synthetic and real-world datasets on 3D object detection. We then validate the design choices via ablation studies.

### 4.1 EXPERIMENTAL SETTINGS AND IMPLEMENTATION DETAILS

We experimented Gaussian-Det on the officially-released benchmark by (Hu et al., 2023) which contain both the synthetic **3D-FRONT** (Fu et al., 2021) and the real-world **ScanNet** (Dai et al., 2017) datasets. We faithfully followed (Hu et al., 2023) to take the posed images as input. The camera poses in the datasets released by (Hu et al., 2023) were estimated via Structure from Motion (SfM) which used the groundtruth point cloud as initialization.

The synthetic **3D-FRONT** (Fu et al., 2021) dataset is featured with its large-scale room layouts and textured furniture models. The dataset contains 159 usable rooms that are manually selected, cleaned and rendered by (Hu et al., 2023). The indoor scenes are densely annotated with oriented bounding boxes. We follow (Hu et al., 2023) to remove the annotations of construction objects such as ceilings and floors, and merge the components that belong to the same object instance into a whole. Please refer to the Appendix pages for illustrative examples of this dataset.

The real-world **ScanNet** (Dai et al., 2017) dataset consists of complex and challenging scenes captured by Structure Sensor in an ego-centric manner. We followed (Hu et al., 2023) to uniformly divide the frames into 100 bins and choose the sharpest one in each bin measured by Laplacian variance. Experiments were done on the 90 scenes that are randomly selected by (Hu et al., 2023).

For the input Gaussians $\mathbb{G}$, we used the official implementation of 3D-GS (Kerbl et al., 2023) equipped with the SuGaR regularization (Guédon and Lepetit, 2024), which takes 30,000 training iterations. We excluded the Gaussians outside the room cuboid defined by (Hu et al., 2023) and filtered out

Table 1: Quantitative results on the 3D-FRONT and ScanNet datasets. †: we report the results of NeRF-MAE which involves the pre-training on 3D-FRONT and two other exterior databases, including HM3D (Ramakrishnan et al., 2021) and Hypersim (Roberts et al., 2021).

| Methods | 3D-FRONT | | | | ScanNet | | | |
|---|---|---|---|---|---|---|---|---|
| | $AR_{25}$ | $AR_{50}$ | $AP_{25}$ | $AP_{50}$ | $AR_{25}$ | $AR_{50}$ | $AP_{25}$ | $AP_{50}$ |
| ImVoxelNet(Rukhovich et al., 2022b) | 88.3 | 71.5 | 86.1 | 66.4 | 51.7 | 20.2 | 37.3 | 9.8 |
| NeRF-RPN (Hu et al., 2023) | 96.3 | 69.9 | 85.2 | 59.9 | 89.2 | 42.9 | 55.5 | 18.4 |
| NeRF-MAE† (Irshad et al., 2024) | 97.2 | 74.5 | 85.3 | 63.0 | **92.0** | 39.5 | 57.1 | 17.0 |
| G-VoteNet (Qi et al., 2019) | 81.5 | 61.6 | 73.0 | 49.6 | 78.5 | 34.2 | 66.8 | 18.2 |
| G-GroupFree (Liu et al., 2021) | 84.9 | 63.7 | 72.1 | 45.1 | 75.2 | 37.6 | 60.1 | 20.4 |
| G-FCAF3D (Rukhovich et al., 2022a) | 89.1 | 56.9 | 73.1 | 35.2 | 90.2 | 42.4 | 63.7 | 18.5 |
| G-BRNet (Xu et al., 2022b) | 89.7 | 75.3 | 88.2 | 71.0 | 71.1 | 32.2 | 63.1 | 19.3 |
| Gaussian-Det (Ours) | **97.9** | **82.3** | **96.7** | **77.7** | 87.3 | **43.0** | **71.7** | **24.5** |

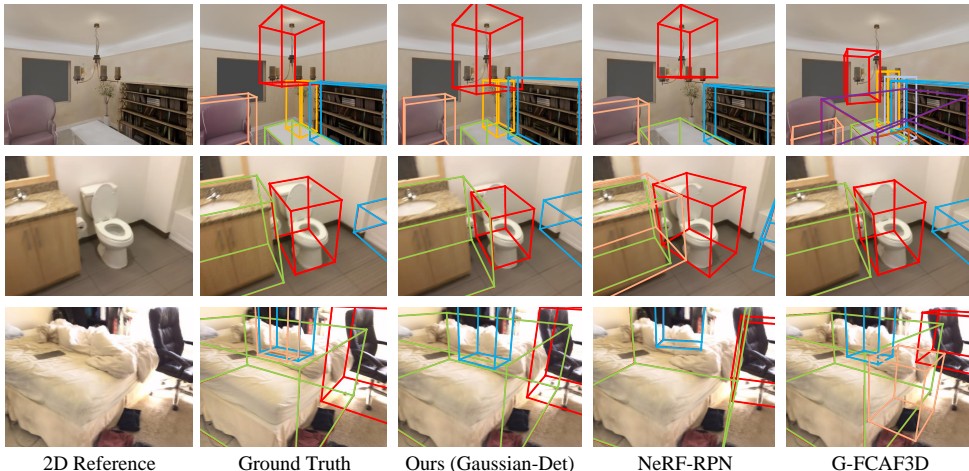

| 2D Reference | Ground Truth | Ours (Gaussian-Det) | NeRF-RPN | G-FCAF3D |

Figure 4: Qualitative Results on 3D-FRONT (top two rows) and ScanNet (bottom two rows). We visualize each bounding box with a unique color for clear illustration.

Gaussians with opacity value lower than 0.3. Note that such a combined implementation (Kerbl et al., 2023; Guédon and Lepetit, 2024) aims at decent 2D rendering results. The issue of outlier Gaussians around the 3D objects still severely exists as shown in Figure 1. We employed PointNet++ (Qi et al., 2017b) as the backbone $\mathcal{B}$. We used a single NVIDIA RTX A6000 48GB GPU for both training and evaluation. The total training epochs of detection is 50 for both 3D-FRONT and ScanNet. The evaluation metrics are mainly average recall (AR) and average precision (AP), thresholded by IoU value at both 0.25 and 0.5. Please refer to the Appendix pages for the details on model hyper-parameters.

From the statistical results (see Appendix) of our reconstructed 3D Gaussians on two datasets, we observe that the majority of flux values are clustered near zero. This indicates that most objects contain a higher degree of surface closure and our assumption is applicable to the employed datasets.

## 4.2 MAIN RESULTS

**Compared Methods:** We have summarized the quantitative results on 3D-FRONT and ScanNet in Table 1. For the comparison with the pure multi-view based methods, we follow (Irshad et al., 2024) to include ImVoxelNet (Rukhovich et al., 2022b), NeRF-RPN (Hu et al., 2023) and NeRF-MAE (Irshad et al., 2024) in Table 1. ImVoxelNet directly extracts 3D volumetric features projected from 2D image planes. The NeRF-based methods (Hu et al., 2023; Irshad et al., 2024) apply 3D convolutions on the uniformly sampled points in the radiance fields. Note that NeRF-MAE includes the pre-training on extra datasets. Moreover, to demonstrate the effectiveness of Gaussian-Det more intuitively, we experimented on the point cloud based detectors (Qi et al., 2019; Liu et al., 2021; Rukhovich et al., 2022a; Cheng et al., 2021) (prefixed by "G-") which merely contains the backbone and detection

Figure 6: Illustration on the detected objects and the surface closure calculated from the 3D Gaussians within each estimated bounding boxes, measured by the absolute flux values (in square decimetre).

heads in Section 3.2. The upper part of Table 1 are cited from (Irshad et al., 2024), for which we have provided a more in-depth comparison in the Appendix pages.

**Analysis on Quantitative Results:** The observation from Table 1 is two-fold. Firstly, the point cloud based methods are able to achieve comparable performances compared with previous multi-view (Rukhovich et al., 2022b) or state-of-the-art NeRF-based (Irshad et al., 2024; Hu et al., 2023) approaches. This suggests the potential of Gaussian Splatting as an explicit input modality for 3D object detection. Secondly, except for AR thresholded by 0.25 on ScanNet, we can see that Gaussian-Det precedes all the compared methods on all evaluation metrics, especially on AP which is more widely-used in 3D object detection. For example, on 3D-FRONT evaluated by $AP_{25}$, Gaussian-Det outperforms the second-best G-BRNet (Cheng et al., 2021) by a significant margin of $+8.5\%$. Although ScanNet is a challenging dataset for its lower image quality and diverse layouts, our Gaussian-Det still achieves the state-of-the-art average precisions ($+4.9\%$ on $AP_{25}$ and $+4.1\%$ on $AP_{50}$). This indicates that Gaussian-Det better leverages the surface-based information inherently condensed in Gaussians. By coalescing partial surfaces into object-wise and closure-aware holistic representations, Gaussian-Det better perceives the quality and reliability of the object proposals, thereby providing favourable information basis for proposal refinement.

**Qualitative Results:** As for illustrative comparison, we firstly display the qualitative comparisons in Figure 4, where we set the 2D rendered results of the input Gaussians as visual reference. We can see that compared with ground truth bounding boxes, Gaussian-Det is capable of preventing redundant object predictions that are potentially caused by outliers, leading to higher detection accuracy. Figure 6 provides a more in-depth visualization on the discriminative objectness deduction of Gaussian-Det. For each detected bounding box, we have marked the corresponding flux value computed in Eqn. (9) that measures the surface closure. Predictions whose absolute flux values are larger or less than one are marked with blue or red respectively. We can see that in the presented examples, a favourable detection well corresponds to a relatively higher degree of closure (lower absolute value of flux). With the holistic surface coalescence by CIM, Gaussian-Det is aware of the reliability of proposals and able to refine them for higher accuracy. Please refer to the Appendix pages for more qualitative results under challenging scenarios.

**Failure Case:** We then present the failure case by Gaussian-Det in Figure 5. Despite focusing on more informative surface representations, Gaussian-Det may still struggle at handling detecting the object with severe occlusion, which is an inherent challenge for current multi-view based 3D object detectors. Moreover, since the surfaces of the objectness that contains jointed objects can also be regarded as closed, Gaussian-Det may mistake them as a whole. Please refer to the Appendix for more failure cases.

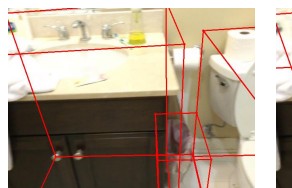 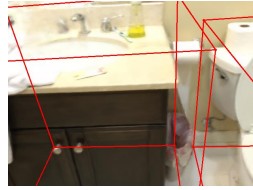

Ground Truth  Ours (Gaussian-Det)

Figure 5: Failure case on occluded objects.

### 4.3 ABLATION STUDIES AND ANALYSIS

To evaluate the effectiveness of the design choices and the components in our framework, we conducted five groups of ablational experiments on the 3D-FRONT dataset. The performances evaluated by average precision (AP) are summarized in Table 2.

(1)-(2) **Discarded or Naive Holistic Surface Coalescence:** The trial (1) aims at experimenting with information on partial surfaces only. This is achieved by concatenating two copies of $\mathbf{f}_k^{part}$ in Eqn. (11). In trial (2), the partial surfaces are coalesced without any consideration on the closure

properties of objects. This is achieved by setting the closure score $S_k = 1$ for all object proposals. By comparing (1), (2) and (6), we can see that the coalescence into holistic surface representation is necessary (+0.7% on $AP_{25}$ and +1.1% on $AP_{50}$) despite its naive implementation in (2). The closure-based measurement augments the proposal representation with prior on the reliability information, which furtherly enhances the detection accuracy sizably and is reflected by the performance gains (+3.2% on $AP_{25}$ and +2.7% on $AP_{50}$).

(3) **Deterministic Partial Surface Representation:** In Gaussian-Det, the feature representation of the trimmed Gaussian candidates are supplemented with a probabilistic residual in Eqn. (6). Here we ignored the variational inference of $\mathcal{F}^{cand}$ by setting $\lambda_{res} = 0$. We can see that the deterministic partial surface representation in (3) is hampered by the under-determined nature of Gaussian Splatting, which leads to inferior performances (-1.0% on $AP_{25}$ and -3.0% on $AP_{50}$).

Table 2: Quantitative results of ablation studies on 3D-FRONT.

|  | $AP_{25}$ | $AP_{50}$ |
|---|---|---|
| (1) Discard the Holistic Surface Coalescence | 92.8 | 73.9 |
| (2) Naive Holistic Surface Coalescence | 93.5 | 75.0 |
| (3) Deterministic Partial Surface Representation | 95.7 | 74.7 |
| (4) No Flexibility in Closure Measurement | 96.4 | 76.2 |
| (5) Alternative Surface Orientation | 96.2 | 75.7 |
| **(6) The Full Framework** | **96.7** | **77.7** |

(4) **No Flexibility in Closure Measurement:** The trial cancels the allowance for the potential truncation in indoor scenes. We empirically set $\gamma_k = 0.5$ for all object proposals. It can be seen that despite the failure case in Figure 5, a learnable $\gamma_k$ endows the closure measurement with the consideration on flexible cases such as truncations and occlusions. Improvements on both $AP_{25}$ (+0.3%) and $AP_{50}$ (+1.5%) can be observed.

(5) **Alternative Surface Orientation:** By default we set the $\mathbf{n}_p$ of a partial surface by solving the eigenvalue problem and determine its in-ward or out-ward by setting the orientation of $\mathbf{n}_p$ to align with the orientation from the center of the object proposal to the Gaussian mean. An alternative way is to determine orientation following (Ran et al., 2022). We experimented on two different implementation of closure scores. As it influences the surface-based Gaussian representation as well as the closure computation, a sizable drops on $AP_{50}$ (-2.0%) can be observed in Table 2.

Please refer to the Appendix pages for additional ablational experiments, including the choices of hyper-parameters, Gaussian Splatting representation and the comparison on learning curves before and after incorporating the probabilistic feature residual.

**Model Efficiency:** As shown in Table 3, Gaussian-Det exhibits remarkable inference speed at 33.3 FPS, which is an order of magnitude faster than both NeRF-RPN and NeRF-MAE. Additionally, our model size is substantially smaller at 24.0MB, less than 8% of NeRF-RPN size and 4% of NeRF-MAE. These results indicate that our Gaussian-Det demonstrates significant improvements in detection accuracy while maintaining real-time processing speed and favourable model compactness.

**Detecting with Noised Poses:** To furtherly validate on inferring the objectness from low-quality data, we experimented on adding normally distributed perturbations with the mean of 0 and variance of 0.07 to camera extrinsics when reconstructing them. The corresponding quantitative results are shown in Table 5. We can see that with noised poses, Gaussian-Det showcases less precision decline on both AP and AR, which verifies the effectiveness of Gaussian-Det against either inherent outlier 3D Gaussians or low-quality input poses. Please refer to the Appendix for qualitative comparisons.

**Efficacy on 3D Instance Segmentation:** We have further validated the generality of our proposed method on the task of open-vocabulary 3D instance segmentation. On the Gaussian-Grouping baseline (Ye et al., 2024), we incorporated the estimation of surface closure as a prior on the reliability of the object instance, thus supporting or suppressing the segmented results. From the results in Table 6, we can see that leveraging the surface closure provides informative prior on objectness deduction and largely enhances the performances of instance segmentation in 3D scenes. The rendering results of qualitative examples containing multiple object instances are shown in Figure 7. It can be seen that the incorporation of the prior of surface closure contributes to fewer noisy predictions off the table. Moreover, it substantially speeds up the training period and saves the GPU memory footprint (see Appendix) compared with the baseline method.

**Comparison with Point-based Partial Surface Modeling:** We note that RepSurf (Ran et al., 2022), a point cloud based convolutional operator, also considers the surface structure within local regions.

Table 3: Comparison on model efficiency.

| Method | FPS | Model Size |
|---|---|---|
| NeRF-RPN (Hu et al., 2023) | 1.51 | 312.7MB |
| NeRF-MAE (Irshad et al., 2024) | 1.11 | 604.2MB |
| Gaussian-Det (Ours) | **33.3** | **24.0MB** |

Table 4: Comparison with RepSurf.

| Method | $AP_{25}$ | $AP_{50}$ |
|---|---|---|
| G-BRNet (Xu et al., 2022b) | 63.1 | 19.3 |
| + RepSurf-U (Ran et al., 2022) | 47.7 | 19.7 |
| Gaussian-Det (Ours) | **71.7** | **24.5** |

Table 5: Comparison under the setting of adding noises to camera poses.

| Methods | Noised Poses | $AR_{25}$ | $AP_{25}$ |
|---|---|---|---|
| G-BRNet (Cheng et al., 2021) | | 89.7 | 88.2 |
| G-BRNet (Cheng et al., 2021) | ✓ | 74.3 (-15.4) | 67.6 (-20.6) |
| Gaussian-Det (Ours) | | 97.9 | 96.7 |
| Gaussian-Det (Ours) | ✓ | 86.8 (**-11.1**) | 81.5 (**-15.2**) |

Table 6: Comparison on 3D instance segmentation.

| Methods | mIoU | mBIoU |
|---|---|---|
| DEVA (Cheng et al., 2023) | 46.2 | 45.1 |
| LERF (Kerr et al., 2023) | 33.5 | 30.6 |
| SA3D (Cen et al., 2023) | 24.9 | 23.8 |
| LangSplat (Qin et al., 2024) | 52.8 | 50.5 |
| Gaussian-Grouping (Ye et al., 2024) | 69.7 | 67.9 |
| Gaussian-Grouping + Ours | **76.5 (+6.8)** | **73.3 (+5.4)** |

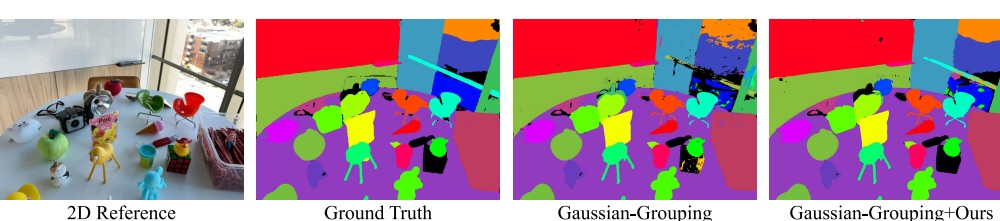

| 2D Reference | Ground Truth | Gaussian-Grouping | Gaussian-Grouping+Ours |

Figure 7: Rendered qualitative results of open-vocabulary 3D instance segmentation.

We accordingly replaced the backbone of G-BRNet (Cheng et al., 2021) with the umbrella variant of RepSurf. The results on ScanNet are shown in Table 4. RepSurf is prone to be severely distracted by the outlier Gaussians since it only considers the very local surfaces, which harms the detection quality drastically (-15.34% on $AP_{25}$). Secondly, merely focusing on the very local (umbrella-shaped in this case) surface while ignoring the holistic surface modeling leads to sub-optimal results (-4.8% on $AP_{50}$). This indicates that the comprehensive surface modeling strategy better captures the intricate geometry of objects, thereby contributing to a sizably improved 3D detection accuracy. Such awareness of holistic surface representation is essential to prevent the distraction by outliers.

## 5 CONCLUSION

In this paper, we present Gaussian-Det for multi-view 3D object detection. Gaussian-Det leverages Gaussian Splatting and exploits the continuous surface representation. To overcome the numerous outliers inherently introduced by the data modality, we devise a Closure Inferring Module (CIM) for comprehensive surface-based objectness deduction. CIM learns the probabilistic feature residual for partial surfaces and coalesces them into holistic representation on closure measurement, thereby serving as a prior on proposal reliability. Experimental results on both synthetic and real-world datasets validate the superiority of our Gaussian-Det over various existing mult-view based approaches.

**Broader Impact:** The proposed method utilizes continuous surface representations to enhance the applications like augmented reality and robotic navigation. While it offers cost-effective solutions via the use of RGB cameras instead of expensive lidar systems, it is largely affected by the quality of input data and may struggle at handling occluded objects. It also raises privacy concerns due to the demand for surveillance devices. To mitigate these issues, the adoption of ethical guidelines is crucial to ensure responsible use and maintain the reliability of the technology.

**Limitations:** Gaussian-Det may be struggling to handle occlusion scenes (as shown in Figure 5), which is an inherent challenge in the setting of multi-view 3D object detection. Specifically, the detectors are unaware of the appearances and detailed geometries of the occluded part, thereby further worsening the inherent ill-posedness of 3D perception given 2D images.

**Future Works:** Since the degree of surface closure of the objectness that contains joint objects is also relatively high, such scenarios might be challenging for Gaussian-Det. Future work can incorporate the color information to discriminate the objects and calculate the flux values separately. In addition, the exploration of large-scale datasets on autonomous driving is a promising practice.

**Acknowledgment:** This work was supported in part by the National Natural Science Foundation of China under Grant 62206147, and in part by the China Postdoctoral Science Foundation under Grant 2024M761674.

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

## A  ADDITIONAL DETAILS ON THE DATASETS

### A.1  DATASET OVERVIEW

In Figure 8, we visualize the scenes in the 3D-FRONT dataset with ground-truth bounding boxes. Each column shows two different views of the same scene. A total of 159 usable rooms are manually selected, cleaned, and rendered to ensure have high quality for scene reconstruction and object detection, as done by NeRF-RPN (Hu et al., 2023).

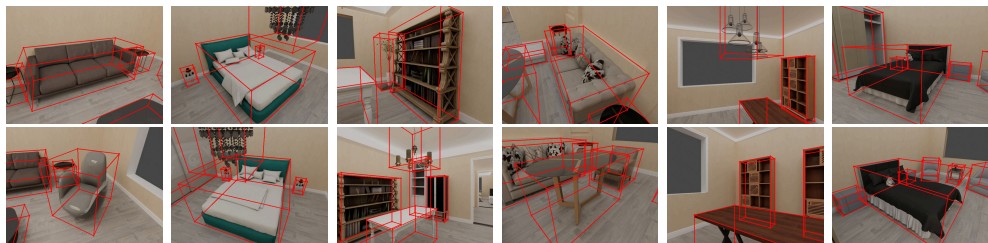

Figure 8: Snapshots of scenes in 3D-FRONT and the corresponding 3D bounding box annotations.

### A.2  STATISTICAL DETAILS

In Figure 9, we have illustrated the statistics on the surface closure of two datasets measured by $|\Phi|$. The bar charts show that the majority of the flux values are clustered near zero, which indicates that most objects in both datasets contain a higher degree of surface closure and our assumption is applicable to the employed datasets.

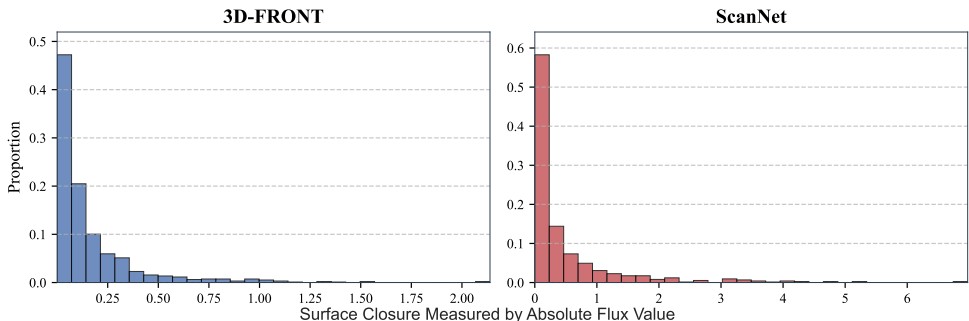

Figure 9: Statistics on surface closure of two datasets (Fu et al., 2021; Dai et al., 2017). Most objects in both datasets contain a higher degree of surface closure. The statistics are collected from our reconstructed 3D Gaussians via 3DGS+SuGAR.

## B  ADDITIONAL TECHNICAL DETAILS

### B.1  MORE DETAILS ON NETWORK ARCHITECTURE

For corresponding feature $F^{cand}$ generation, we use the same PointNet++ backbone in VoteNet (Qi et al., 2019), which is the backbone $\mathcal{B}$ in Eqn. 4. Specifically, it has four set abstraction (SA) layers and two feature propagation/upsampling (FP) layers. SA layers sub-sample the input to 2048, 1024, 512 and 256 Gaussians respectively. The two FP layers up-sample the 4th SA layer's output back to 1024 Gaussians with 256-dim features and 3D coordinates.

The querying-and-pooling function in Eqn. 5 and Eqn. 10 has the output size of 128. Therefore, $f^{part}$ and $f^{hol}$ have the same feature channels of 128 and $f^{final}$ has the channel of 256.

The stacked MLP for proposal refinement in Eqn. 12 has output size of 128, 128, 9. In these 9 channels, the first two are for objectness classification, the following one is for heading angle refinement and the last six are for offsets refinement. The initial proposal has the same shape with refined proposal and is generated in the same way.

For the point cloud based methods, the original 3D Gaussians $\mathbb{G} = \{\mathbf{g}_p = (\mu_p, \mathbf{\Sigma}_p, \alpha_p, \mathbf{c}_p)\}_p^P$ (generated by 3DGS+SuGAR) were used as the input, where the Gaussian center $\mu_p$ was regarded as the input point-wise coordinate. $\mathbf{\Sigma}_p, \alpha_p, \mathbf{c}_p$ were concatenated and regarded as the input point-wise features. Except for the first layer whose input dimension was modified to fit the input Gaussians, we faithfully followed the network and training configurations of the point cloud based methods (Qi et al., 2019; Liu et al., 2021; Rukhovich et al., 2022a; Xu et al., 2022b) we compared with.

## B.2 Details on Calculating Surface Normal

We provide details of calculating surface normal, which is determined by solving the eigenvalue problem:

$$\mathbf{\Sigma_p n_p} = \lambda \mathbf{n_p} \tag{14}$$

where $\mathbf{\Sigma}_p = \mathbf{R}_p \mathbf{S}_p \mathbf{S}_p^T \mathbf{R}_p^T$. Then we made identical deformation to both sides of the equation.

$$\mathbf{R}_p \mathbf{S}_p \mathbf{S}_p^T \mathbf{R}_p^T \mathbf{n}_p = \lambda \mathbf{n}_p$$
$$\mathbf{S}_p^T \mathbf{R}_p^T \mathbf{R}_p \mathbf{S}_p \mathbf{S}_p^T \mathbf{R}_p^T \mathbf{n}_p = \lambda \mathbf{S}_p^T \mathbf{R}_p^T \mathbf{n}_p \tag{15}$$

We set $\mathbf{S}_p^T \mathbf{R}_p^T \mathbf{n} = \mathbf{x}$. As $\mathbf{R}_p$ is a rotation matrix, we get $\mathbf{R}_p^T \mathbf{R}_p = 1$. Then:

$$\mathbf{S}_p^T \mathbf{S}_p \mathbf{x} = \lambda \mathbf{x} \tag{16}$$

The solution is $\mathbf{x}_0 = [1, 0, 0]^T, \mathbf{x}_1 = [0, 1, 0]^T, \mathbf{x}_2 = [0, 0, 1]^T$. Therefore:

$$\mathbf{n}_{p,i} = \mathbf{R}_p \mathbf{S}_p^{-1} \mathbf{x}_i, i = 0, 1, 2 \tag{17}$$

We define eigenvector with smallest eigenvalue as the **Surface Normal** $\mathbf{n}_p$. The defination is consistent with (Guédon and Lepetit, 2024). However, $\mathbf{n}_p$ is unoriented, meaning it can point either inside or outside. Therefore, we determine the surface normal direction by considering the object center (in practice, we use proposal center). To elaborate, the orientation of surface normal will align with the orientation pointing from the object center to the Gaussian center. An alternative strategy is to follow (Ran et al., 2022) to determine the orientation. Specifically, we ensure that the first element of the normal vector is non-negative. Then we determine the orientation using an instance-level random inversion with $p = 50\%$. Ablation study in Table 2 shows that the result of strategy of (Ran et al., 2022) is lower than the aforementioned strategy.

## B.3 More Details on Training

In training stage, we optimize the residue loss by maximizing the evidence lower bound.

$$\mathcal{L}_{res} = \mathcal{L}_{\boldsymbol{\mu_g}, \sigma_\mathbf{g}}(\mathcal{F}^{cand}) = \mathbb{E}_{q(\mathcal{F}^V | \mathcal{F}^{cand})}[\log p(\mathcal{F}^{cand} | \mathcal{F}^V)] - \text{KL}(q(\mathcal{F}^V | \mathcal{F}^{cand}) || p(\mathcal{F}^V)) \tag{18}$$

After applying the reparameterization trick (Bengio et al., 2013), the total variational loss can be reduced to:

$$\mathcal{L}_{res} = \approx \frac{1}{M} \sum_{i=1}^{M} \log p_\theta(\mathcal{F}^{cand} | \mathcal{F}^V)$$
$$+ \frac{1}{2M} \sum_{i=1}^{M} \sum_{j=1}^{C} \left(1 + \log((\boldsymbol{\sigma}_{\mathbf{g}_i}^{(j)})^2) - (\boldsymbol{\mu}_{\mathbf{g}_i}^{(j)})^2 - (\boldsymbol{\sigma}_{\mathbf{g}_i}^{(j)})^2\right) \tag{19}$$
$$\triangleq \mathcal{L}_{rec} + \mathcal{L}_{appr}$$

where $C$ refers to channel and $M$ refers to the number of candidate Gaussians. $\mathcal{L}_{rec}$ refers to reconstruction loss and $\mathcal{L}_{appr}$ refers to Gaussian approximation loss. During training we implement square difference for simplicity.

## B.4 Proof of Theorem 1

Theorem 1 is a special case of Gauss' Flux Theorem by setting $\mathbf{T}$ as any constant vector field:

$$\oiint_S \mathbf{T} \cdot \mathbf{n} dS = \iiint_V \nabla \cdot \mathbf{T} dV \overset{\nabla \cdot \mathbf{T} = 0}{=} 0 \tag{20}$$

Table 7: The hyper-parameter setup in Gaussian-Det.

| Name | Description | Value |
|---|---|---|
| $\lambda_{res}$ | Weight of $\mathcal{L}_{res}$ | 1 |
| $\lambda_{pred}$ | Weight of $\mathcal{L}_{pred}$ | 1 |
| $\lambda_{refine}$ | Weight of $\mathcal{L}_{refine}$ | 1 |
| $\alpha$ | Residue weight | 0.1 |
| $\delta$ | Threshold of grouping the Gaussians | 0.2 |
| $P$ | The number of input Gaussians | 40000 |
| $M$ | The number of candidate Gaussians $\mathbb{G}^{cand}$ | 1024 |
| $C$ | Channel of candidate feature $\mathcal{F}^{cand}$ | 256 |

## C  ADDITIONAL ABLATION EXPERIMENTS AND ANALYSIS

### C.1  HYPER-PARAMETER SETUP

We summarize the hyper-parameters used in Table 7. These include hyper-parameters for loss function, grouping threshold, input and network setup. We have also tried different choices of $\lambda_{res}$. We choose $\lambda_{res} = 1$ as the default setting.

Table 8: Experimental results of parameter settings on Gaussian-Det. $\lambda_{res} = 0$ denotes deterministic partial surface representation in ablation study. Performances of detection on 3D-FRONT are reported.

| $\lambda_{res}$ | Is Default | AP$_{25}$ | AP$_{50}$ |
|---|---|---|---|
| 0 | | 95.7 | 74.7 |
| 0.1 | | 95.5 | 76.1 |
| 0.5 | | 96.2 | 76.6 |
| 1 | ✓ | 96.7 | 77.7 |
| 2 | | 96.2 | 75.7 |

We have also collected the statistics before and after removing $F_V$ and illustrated the learning curve in Figure 10. We can observe that the probabilistic feature residual term contributed to a slightly faster initial convergence and less fluctuation throughout the training process. This suggests that Gaussian-Det with $F_V$ is more stable during training.

### C.2  IMPACTS OF $F_V$ ON THE TRAINING PROCESS

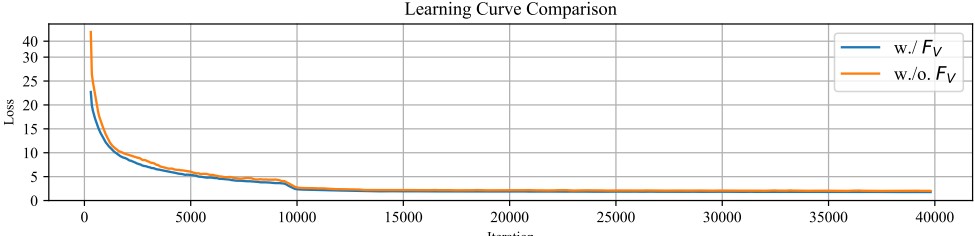

Figure 10: Training curves before and after removing $F_V$.

### C.3  ANALYSIS ON UNDERLYING GAUSSIAN REPRESENTATION

To investigate how the underlying Gaussian representation, we have experimented on the ablation study by detecting from 3G Gaussians that were reconstructed via the original 3DGS (Kerbl et al., 2023) algorithm.

The quantitative results are displayed in Table B. While the detection accuracies drop with the input of lower quality 3D Gaussians, they still outnumber all the compared methods in Table 9 which consume higher-quality 3D Gaussians that were reconstructed via 3DGS+SuGaR. We additionally show experimental results using 2DGS for better surface modeling quality and improvements can be observed.

Table 9: Quantitative results on the Gaussian representation.

| Methods | 3D-FRONT | | ScanNet | |
|---|---|---|---|---|
| | $AP_{25}$ | $AP_{50}$ | $AP_{25}$ | $AP_{50}$ |
| Gaussian-Det (3DGS+SuGaR (Kerbl et al., 2023; Guédon and Lepetit, 2024)) | 96.7 | 77.7 | 71.7 | 24.5 |
| Gaussian-Det (3DGS (Kerbl et al., 2023)) | 94.8 | 74.3 | 68.9 | 22.0 |
| Gaussian-Det (2DGS (Huang et al., 2024)) | 95.3 | 79.9 | 70.1 | 24.6 |

# D   ADDITIONAL VISULIZATION RESULTS

## D.1   ADDITIONAL QUALITATIVE RESULTS ON ABLATION STUDIES

In Figure 11, we have accordingly illustrated the qualitative comparisons of the ablation studies on different components (see Figure 11 in the Appendix of the revised manuscript). It can be seen that either discarding the holistic surface coalescence or using the deterministic partial surface representation leads to sub-optimal visual results, such as inaccurate sizes and locations.

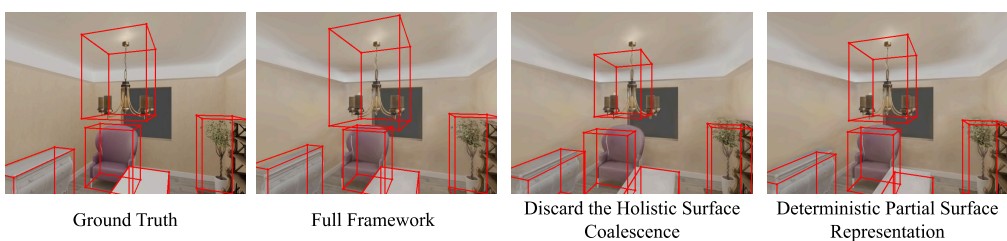

Ground Truth        Full Framework        Discard the Holistic Surface Coalescence        Deterministic Partial Surface Representation

Figure 11: Additional qualitative results on ablation studies.

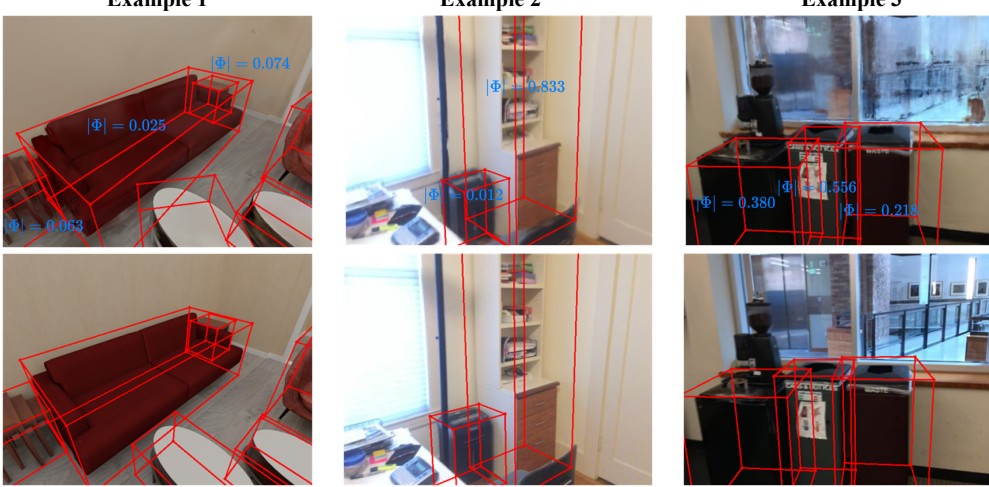

Figure 12: Detection results (upper) under challenging scenarios and corresponding ground-truths (bottom).

## D.2   ADDITIONAL QUALITATIVE RESULTS UNDER CHALLENGING SCENARIOS

Apart from the failure case in Figure 5, we illustrate another case in Example 1, Figure 12 where Gaussian-Det successfully detects the occluded object. In this example, Gaussian-Det benefits from the prior that the objectness contains a high degree of surface closure ($|\Phi| = 0.074$), thereby tackling the occlusion caused by the sofa ($|\Phi| = 0.025$).

On the other hand, for those challenging objects with higher that take up a small proportion, we have illustrated similar cases of ScanNet in Example 2&3 of Figure 12. In Example 2, a bookshelf ($|\Phi| = 0.833$) is close to a suitcase and two walls. In Example 3, a desk ($|\Phi| = 0.380$) and two garbage bins ($|\Phi| = 0.556, 0.218$) are tightly positioned to each other and are against a wall. The absolute flux values ($|\Phi|$) of these exemplified objects are relatively higher due to less exposure to

camera shooting. While our assumption is weaker in these examples, our Gaussian-Det is still able to detect the target objects.

In Figure 13, we have included more qualitative results on occlusions, where the compared methods predict inaccurate sizes or locations.

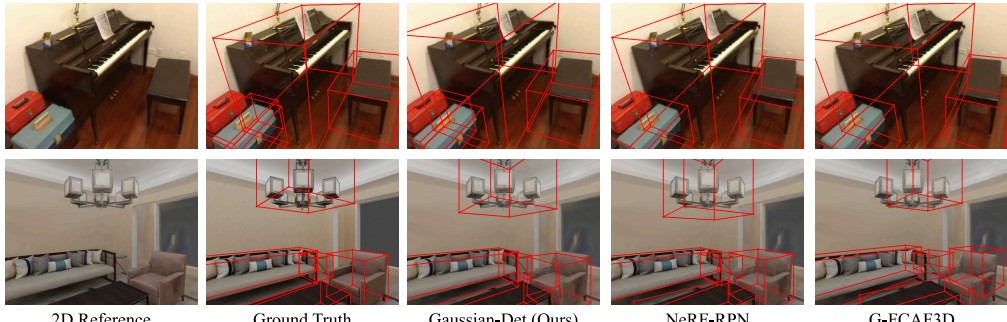

|  |  |  |  |  |
| --- | --- | --- | --- | --- |
| 2D Reference | Ground Truth | Gaussian-Det (Ours) | NeRF-RPN | G-FCAF3D |

Figure 13: Additional qualitative results on challenging cases and comparisons with other methods.

Another challenging scenario is detecting from low-quality 3D Gaussians that are reconstructed from images with noised poses. From Figure 14, we can see that Gaussian-Det is still capable of detecting the target objects, especially the table which is visually blurred after rendering.

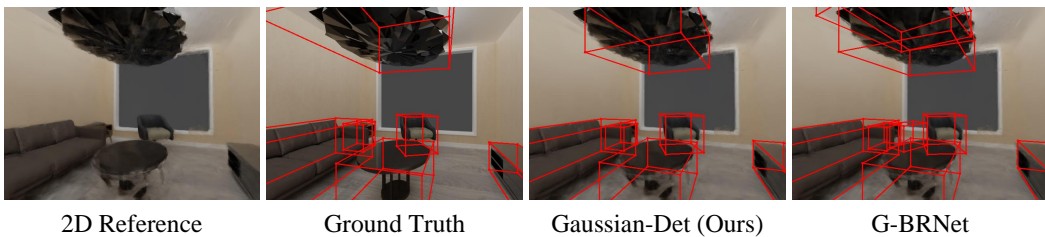

|  |  |  |  |
| --- | --- | --- | --- |
| 2D Reference | Ground Truth | Gaussian-Det (Ours) | G-BRNet |

Figure 14: Qualitative results on low-quality Gaussians reconstructed from images with noised poses.

### D.3    ADDITIONAL FAILURE CASES

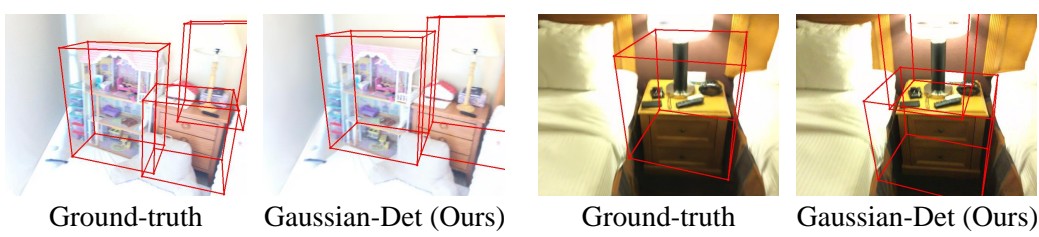

|  |  |  |  |
| --- | --- | --- | --- |
| Ground-truth | Gaussian-Det (Ours) | Ground-truth | Gaussian-Det (Ours) |

Figure 15: Additional failure cases.

Apart from the example provided, we have further presented more failure cases in Figure 15. Since the surfaces of the objectness that contains jointed objects can also be regarded as closed, Gaussian-Det may mistake them as a whole. Future researches can focus on incorporating the color information to discriminate joint objects.

## E    ADDITIONAL QUANTITATIVE RESULTS

### E.1    DETAILED COMPARISON WITH NERF-RPN AND NERF-MAE

In Table 10, we report all the available officially-reported results of NeRF-RPN and NeRF-MAE for a thorough comparison with Gaussian-Det. The checkmark indicates that NeRF-MAE employs self-supervised pre-training before finetuning on the downstream detection task. F3D, HM3D and

Table 10: Detailed quantitative comparison on 3D-FRONT and ScanNet.

| Reference | Methods | Pretrained | 3D-FRONT | | | | ScanNet | | | |
|---|---|---|---|---|---|---|---|---|---|---|
| | | | $AR_{25}$ | $AR_{50}$ | $AP_{25}$ | $AP_{50}$ | $AR_{25}$ | $AR_{50}$ | $AP_{25}$ | $AP_{50}$ |
| (Hu et al., 2023) | NeRF-RPN (AB, VGG19) | | 97.8 | 76.5 | 65.9 | 43.2 | 88.7 | 42.4 | 40.7 | 14.4 |
| | NeRF-RPN (AB, R50) | | 96.3 | 70.6 | 65.7 | 45.1 | 86.2 | 32.0 | 34.4 | 9.0 |
| | NeRF-RPN (AB, Swin-S) | | 98.5 | 63.2 | 51.8 | 26.2 | 93.6 | 44.3 | 38.7 | 12.9 |
| | NeRF-RPN (AF, VGG19) | | 96.3 | 69.9 | 85.2 | 59.9 | 89.2 | 42.9 | 55.5 | 18.4 |
| | NeRF-RPN (AF, R50) | | 95.6 | 67.7 | 83.9 | 55.6 | 91.6 | 35.5 | 55.7 | 16.1 |
| | NeRF-RPN (AF, Swin-S) | | 96.3 | 62.5 | 78.7 | 41.0 | 90.6 | 39.9 | 57.5 | 20.5 |
| (Irshad et al., 2024) | NeRF-RPN | | 96.3 | 69.9 | 85.2 | 59.9 | 89.2 | 42.9 | 55.5 | 18.4 |
| | NeRF-MAE (F3D) | ✓ | 96.2 | 67.5 | 78.0 | 54.3 | 89.7 | 36.1 | 51.0 | 14.5 |
| | NeRF-MAE (F3D, Aug) | ✓ | 96.3 | 74.3 | 83.0 | 59.1 | 90.5 | 39.1 | 54.3 | 15.5 |
| | NeRF-MAE (F3D&HM3D&HS, Aug) | ✓ | 97.2 | 74.5 | 85.3 | 63.0 | 92.0 | 39.5 | 57.1 | 17.0 |
| | Gaussian-Det (Ours) | | 97.9 | 82.3 | 96.7 | 77.7 | 87.3 | 43.0 | 71.7 | 24.5 |

Table 11: Comparison of model efficiency on the task of 3D instance segmentation.

| Methods | Training Time | GPU Memory |
|---|---|---|
| Gaussian-Grouping (Ye et al., 2024) | 1.33h | 35.8GB |
| Gaussian-Grouping + Ours | **0.65h** | **12.8GB** |

HS refer to the 3D-FRONT, HM3D and Hypersim datasets for self-supervised pre-training. Aug. indicates augmentations done during pre-training. Note that without pre-training on exterior databases, Gaussian-Det still outperforms variants of both NeRF-RPN and NeRF-MAE on most evaluation metrics.

### E.2 MODEL EFFICIENCY ON 3D INSTANCE SEGMENTATION

Apart from the quantitative results in Table 6, we have additional measured model efficiency in Table 11. We can see that leveraging the surface closure can substantially speed up the training period and significantly save the GPU memory footprint.

