# OpenReview forum: "Gaussian-Det: Learning Closed-Surface Gaussians for 3D Object Detection"
_ICLR.cc/2025/Conference — ICLR 2025 Poster_

### Official Review · Reviewer_kAE9 · 2024-10-31

**Soundness:** 3
**Presentation:** 2
**Contribution:** 3
**Rating:** 6
**Confidence:** 3

**Summary:**

The paper presents Gaussian-Det, a method to detect 3D objects via Gaussian Splatting. The proposed method leverages Gaussian Splatting as surface representation for multi-view based 3D object detection. Gaussian-Det proposes Closure Inferring Module (CIM) to deal with the outliers present using Gaussian-Splatting technologies. The paper presents experiments on both synthetic and real 3D-object detection datasets where Gaussian-Det mostly outperforms previous methods.

**Strengths:**

- Interesting solution using Gaussian-Splatting technologies for 3D object detection.
- Proposing Closure Inferring Module (CIM) to deal with the noisy nature of Gaussian-Splatting representations.

**Weaknesses:**

- Clarity of the paper is not good and needs improvement (my most concerning weakness). First, overall the high-level idea is not clearly explained. Although, Fig. 2 tries to do this, it falls short. Figure 2 lacks a clear illustration of the goal using the Backbone, and most importantly, fails to explain clearly how the CIM module (in my opinion the main contribution). Second, given the lack of clarity, it is hard to judge the novelty and quality of the solution. For example, I don't find the description of the Theorem if the high-level idea is not explained clearly. In my opinion it is just using space that could've been used to add Figures to explain CIM in a better way.

- Lack of details in the experiments. First, in line 320-321, the narrative says that the scenes are manually downsized into 159 usable rooms. However, it is not clear if the data was verified to ensure high quality. This is important since this data is used to measure quality.

- Insufficient experiments. In my opinion, showing results only on 2 datasets is not that convincing. Given that 3D object detection has been around for some time, it would've been more informative to know the performance of this proposed method on more real-scenes benchmarks/datasets (e.g., Waymo Open datasets, Objectron, etc.). I think that these datasets could've revealed the practical benefit of the proposed approach.

**Questions:**

See Weaknesses section.

---

> ### Author Response · Authors · 2024-11-22
> **Response (Part I)**
>
> We truly appreciate your valuable comments. In the following, we provide responses to the concerns.
>
> **Q1: The goal of using the backbone in the framework figure.**
>
> A: The backbone aims at downsampling the large amount of (usually >100k) 3D Gaussians into a smaller and fixed number (<10k) of candidate Gaussians, while extracting their corresponding features. In the framework figure (Figure 3) of the revised manuscript, we have (1) used an isosceles trapezoid and (2) noted the number of input&output 3D Gaussians to highlight the downsampling process.
>
> **Q2: How the CIM module works in the framework figure.**
>
> A: In the framework figure (Figure 3) of the revised manuscript, we have made the following polishments to better convey how CIM works in Gaussian-Det:
>
> - We have included the text of ''CIM'' into two subheads in the bottom to highlight that they belong to the CIM module.
> - We have followed the commonly-used illustration of the residual connection by adding a shortcut in the part of ''CIM: Partial Surface Inference''.
> - In the part of ''CIM: Holistic Surface Closure Coalescence'', we have incorporated illustrative examples of how CIM uses the surface closure (measured by $|\Phi|$) as the quantitative prior on the object proposals. In the illustrative examples, we have established correspondences between the framework figure (Figure 3) and Figure 2 to better motivate our design choice.
>
> **Q3: Fitting Theorem 1 into the elaboration of CIM module.**
>
> A: Thank you for your insightful suggestion. Theorem 1 mathematically defines our measurement of surface closure. We strongly agree that it is crucial to conceptually introduce such a measurement, and properly convey how the measurement is leveraged by the CIM module in Gaussian-Det. In the revised manuscript, we have re-organized the content in Section 3.3. Specifically, we have made the following polishments to better fit Theorem 1 into the elaboration of CIM module conceptually, mathematically and practically:
>
> - **Before Theorem 1:** We have moved the illustrative example of the surface closure prior ahead, thus bringing up earlier the concept that the surface closure is utilized as a measurement of the quality of the predicted objectness.
> - **Before Theorem 1:** We have highlighted that CIM leverages a key geometric
> property that the partial surfaces corresponding to an object can approximate a closed surface (L259-L262 in the revised manuscript), thereby naturally educing the details of Theorem 1 that mathematically describes this property.
> - **After Theorem 1:** In the following narration, we have noted how to utilize such a property as a measurement of the objectness quality in practice.
> - The proof of Theorem 1 has been moved to the Appendix for brevity.

---

> ### Author Response · Authors · 2024-11-22
> **Response (Part II)**
>
> **Q4: Verification on the quality of 3D-FRONT dataset.**
>
> A: Thank you for pointing this out. In [1], the authors mentioned that ''effort has been spent on splitting complex scenes into individual rooms and cleaning up bounding boxes. A total of 159 usable rooms are manually selected, cleaned, and rendered in 3D-FRONT. ''  Moreover, extensive manual cleaning is conducted on the bounding boxes in each room. We have also checked the image data of this dataset and illustrated the example scenes in the Appendix of the revised manuscript (see Figure 8).
>
> **Q5: Results on extra real-scene dataset.**
>
> A: During the discussion phase, we have further experimented on the task of open-vocabulary instance segmentation in 3D scenes, which detects the target objects in real-world 3D datasets in a much finer granularity. On the Gaussian-Grouping baseline [2], we have implemented the estimation of surface closure as a prior on the reliability of the object instance, thus supporting or suppressing the segmented results.
>
> From the quantitative results in Table D&E (Table 6&11 in the revised manuscript), we can see that leveraging the surface closure provides an informative prior on objectness deduction and largely enhances the performance of instance segmentation in 3D scenes. Moreover, it substantially speeds up the training period and saves the GPU memory footprint after ruling out potential outliers and cutting down the total number of 3D Gaussians. The rendered results of qualitative examples containing multiple object instances are shown in Figure 7 of the revised manuscript. It can be seen that the incorporation of the prior of surface closure contributes to fewer noisy predictions.
>
> We are sorry that during the discussion phase, we are provided with limited time and computational resources to perform the 3D Gaussian Splatting on the large-scale dataset for autonomous driving. This will be our future work where we investigate how Gaussian-Det performs on the outdoor datasets.
>
> ### Table D: Open-vocabulary 3D instance segmentation on the real-world LERF-Mask/figurines dataset
>
>
> | Methods                             | mIoU | mBIoU |
> |-------------------------------------|------|-------|
> | DEVA (ICCV'2023)           | 46.2 | 45.1  |
> | LERF (ICCV'2023)            | 33.5 | 30.6  |
> | SA3D (NeurIPS'2023)             | 24.9 | 23.8  |
> | LangSplat (CVPR'2024)        | 52.8 | 50.5  |
> | Gaussian-Grouping (ECCV'2024) | 69.7 | 67.9  |
> | **Gaussian-Grouping+Ours**             | **76.5** | **73.3**  |
>
> ### Table E: Comparisons on model efficiency in the task of open-vocabulary 3D instance segmentation.
>
> | Methods                             | Training Time | GPU Memory |
> |-------------------------------------|---------------|------------|
> | Gaussian Grouping (ECCV'2024) | 1.33h         | 35.8GB     |
> | **Gaussian-Grouping+Ours**            | **0.65h**     | **12.8GB** |
>
> [1] Hu, Benran, et al. "Nerf-rpn: A general framework for object detection in nerfs." CVPR. 2023.
>
> [2] Ye, Mingqiao, et al. "Gaussian grouping: Segment and edit anything in 3d scenes." ECCV, 2024.

---

> > ### Comment · Reviewer_kAE9 · 2024-11-26
> > **RE: Response**
> >
> > Thanks for providing answers to my concerns. Most of them are addressed. However, if the manuscript eventually gets accepted, please include the clarifications and explanations to make the paper more clear.

---

> > > ### Author Response · Authors · 2024-11-26
> > >
> > > Thank you for your positive feedback. We have incorporated all the suggested clarifications and explanations into current version of the manuscript to enhance its clarity and comprehensiveness. We will also make sure that all these clarifications and explanations above are included in the final version.

---

### Official Review · Reviewer_nYGE · 2024-11-03

**Soundness:** 3
**Presentation:** 1
**Contribution:** 2
**Rating:** 6
**Confidence:** 3

**Summary:**

This paper introduces a novel 3D object detection method, Gaussian-Det, which uses 3DGS to represent objects as continuous surfaces rather than NeRF discrete points. The method uses Gaussian as a surface descriptor and introduces the Closure Inference Module (CIM), which solves the outlier problem inherent in the 3DGS method by considering surface closure as a prior for objectness. The CIM operates in two phases: firstly, the probabilistic feature residuals are estimated for some of the surfaces, and then they are merged to form a holistic representation that measures the overall surface closure. The method is evaluated on synthetic (3D-FRONT) and real-world (ScanNet) datasets, and the results show a significant improvement over previous SOTA methods. The main innovations of the method are its surface-based representation and the use of surface closure as a geometric prior to improve the quality of detection.

**Strengths:**

- The proposed method show SOTA performance compared with previous NeRF based methods.
- Use 3DGS is faster than NeRF.
- The CIM module effectively handles outliers in Gaussian Splatting.

**Weaknesses:**

- There is a lack of qualitative comparisons between different components, particularly regarding the use of Holistic Surface Coalescence and Residual Estimation. These comparisons are important for readers to better understand the effectiveness of each component.
- It would be interesting to analyze how the underlying Gaussian representation affects the results. For example, what is the impact of using the original 3DGS or a 2DGS representation?
- The overall writing is unclear and difficult to follow.
	- L.208 should be "From G^{cand}"
	- There is multiple M in FIg. 2, are they the same?
	- Figure 3 is confusing. It is unclear what “open surfaces” and “closed surfaces” refer to. In Figure 3 (a) and (b), the primary visual difference appears to be the incompleteness of the Gaussians in (b). Please clarify the relationship between open and closed surfaces.

**Questions:**

Improving readability and providing a visual comparison of the ablations would better present this work.

---

> ### Author Response · Authors · 2024-11-22
> **Response**
>
> We truly appreciate your valuable comments. In the following, we provide responses to the concerns.
>
> **Q1: Qualitative comparisons with ablated components.**
>
> A: Thank you for your valuable suggestion. In the revision, we have accordingly illustrated the qualitative comparisons of the ablation studies on different components (see Figure 11 in the Appendix of the revised manuscript). It can be seen that either discarding the holistic surface coalescence or using the deterministic partial surface representation leads to sub-optimal visual results, such as inaccurate sizes and locations.
>
> **Q2: How the underlying Gaussian representation affects the results.**
>
> A: Thank you for your insightful suggestion. To investigate the effect of underlying Gaussian representation, in the discusion phase, we have experimented on the ablation study you mentioned by detecting from 3D Gaussians that were reconstructed via the original 3DGS[1] algorithm.
> The quantitative results are displayed in Table C. While the detection accuracies drop with the input of lower quality 3D Gaussians, they still outperform all the compared methods in Table 1 which consume higher-quality 3D Gaussians that were reconstructed via 3DGS+SuGaR. In the revised manuscript, we have included the results in Table C and the analysis (see Appendix D.1).
>
> ### Table C: Quantitative results on the Gaussian representation.
> |Setting|3D-FRONT (AP25)|3D-FRONT (AP50)|ScanNet (AP25)|ScanNet (AP50)|
> |--|--|--|--|--|
> |Gaussian-Det (3DGS+SuGaR)|96.7|77.7|71.7|24.5|
> |Gaussian-Det (3DGS)|94.8|74.3|68.9|22.0|
>
> **Q3 part 1: Clarifications on the writing of the proposed method**
>
> A: We would like to thank you for your suggestions on improving the overall clarity of the manuscript. In the revision, we have thoroughly gone through the elaboration of our method and made the following improvements:
>
> - **On denotations:** We have checked the math denotations that you have mentioned and corrected the usages.
> - **On network modules:** We have differentiated between the two $M$s and highlighted that they are independent but sharing the same architecture. Moreover, we have used an isosceles trapezoid to represent the backbone in the framework figure (Figure 3 in the revised manuscript), to highlight its downsampling effect.
>
>
> **Q3 part 2: Clarification on relationship between open and closed surfaces:**
>
> A: Ideally, a set of closed surfaces can fully enclose a volume. However, when there exists obvious discontinuity in this enclosure due to incompleteness or the introduction of outlier surfaces, the set of surfaces does not fully enclose a volume and are regarded as open. Such enclosure is quantitatively measured by the flux value in Theorem 1. The more closed a set of surfaces is, the closer the $|\Phi|$ value computed on them is to zero.
>
> We agree that Figure 3 in our original manuscript is crucial for motivating our method and any confusion should be avoided. In the revised manuscript, we have moved it ahead as Figure 2 and clarified that the open or closed surfaces are measured by the $|\Phi|$ value. In the framework figure (Figure 3) of the revised manuscript, we have also added the necessary information that $|\Phi|$ measures how well the surfaces enclose a volume, which is used to discriminate the degree of surface closure.
>
> [1] Kerbl, Bernhard, et al. "3D Gaussian Splatting for Real-Time Radiance Field Rendering." ACM Transactions on Graphics 42.4 (2023): 1-14.

---

> > ### Comment · Reviewer_nYGE · 2024-11-27
> >
> > Thank you for your reply. The reply addressed most of my concerns and I will raise my score. Besides, I suggest authors add an experiment of using 2DGS as the underlying representation, as it shows better surface quality.

---

> > > ### Author Response · Authors · 2024-11-28
> > >
> > > We sincerely appreciate your valuable comments and questions, which have greatly contributed to improving the manuscript. We are also grateful for the increase in your score. We have initiated experiments on 2DGS following your suggestion and plan to incorporate any meaningful findings in future revisions.

---

### Official Review · Reviewer_XhiG · 2024-11-04

**Soundness:** 3
**Presentation:** 3
**Contribution:** 3
**Rating:** 6
**Confidence:** 4

**Summary:**

In this paper is presented a novel method for multi-view 3D object detection, exploiting Gaussian splatting to obtain a continuous scene representation. Then, to handle outliers in the previous representation, a closure inferring module is included that learns a probabilistic feature residual for partial surfaces and coalesces them into holistic representation on closure measurement. Experimental evaluation is reported on both synthetic and real datasets, including an ablation study and some comparisons with respect to competing approaches.

**Strengths:**

The paper is addressing a complex and important problem.

The authors include most of the details to understand their proposal, being the technical section clear enough and well written.

Gaussian-Det is simple and effective. To this end, the overall framework is divided into the construction of surfaces based on Gaussian representations, an object proposal initialization, as well as a partial surface feature inference and holistic surface closure coalescence.

Both synthetic and real experiments are provided in the paper. The results are competitive with respect to several competing approaches.

The ablation study helps the reader.

**Weaknesses:**

- Camera poses exploit a type of ground truth (point cloud) to be inferred. In my opinion, this is clearly a strong limitation of the paper.

- Lack of challenging cases. In my opinion, some qualitative instances could be included in the paper, showing the effectiveness of the method in complex cases, especially, in comparison with other approaches.

**Questions:**

To be honest, I do not have many issues with the current submission.

In figure 6, maybe the authors could use a different color per object.

How could a noisy estimate of the cameras affect overall performance?

---

> ### Author Response · Authors · 2024-11-22
> **Response**
>
> We truly appreciate your valuable comments. In the following, we provide responses to the concerns.
>
> **Q1: Camera poses exploit a type of ground truth (point cloud) to be inferred. In my opinion, this is clearly a strong limitation of the paper. How could a noisy estimate of the cameras affect overall performance?**
>
> A: Thank you for pointing this out. We agree that since the officially-released dataset of [1] uses ground truth point cloud to infer camera poses, the effectiveness of our proposed method can be better reflected under the setting of noised camera parameters. To this end, we have accordingly conducted benchmark experiments with noised poses as input. Specifically, we manually added normally distributed perturbations with mean of 0 and variance of 0.07 to the camera extrinsics estimated by [1]. The quantitative results are shown in Table B. We can see that with noised poses, Gaussian-Det showcases less decline on both AP and AR, which verifies the effectiveness of Gaussian-Det against either inherent outlier 3D Gaussians or low-quality input poses. In the revision, we have included the quantitative comparison shown in Table B into the Experiment section and demonstrated the corresponding qualitative results in the Appendix.
>
> ### Table B: Comparison under Noised Poses
>
> | Methods        | Noised Poses | AR<sub>25</sub> | AP<sub>25</sub> |
> |----------------|--------------|-----------------|-----------------|
> | G-BRNet        |              | 89.7            | 88.2            |
> | G-BRNet        | ✓            | 74.3 (-15.4)   | 67.6 (-20.6)   |
> | **Ours**       |              | 97.9            | 96.7            |
> | **Ours**       | ✓            | 86.8 **(-11.1)**   | 81.5 **(-15.2)**   |
>
>
> **Q2: Lack of challenging cases. In my opinion, some qualitative instances could be included in the paper, showing the effectiveness of the method in complex cases, especially, in comparison with other approaches. In figure 6, maybe the authors could use a different color per object.**
>
> A: Thank you for your valuable suggestion. In the original submission (Figure 9 in Appendix), we have included several qualitative instances under challenging scenarios including occlusions and jointedness. In Figure 13 of the revised manuscript, we have included more qualitative results on occlusions, where the compared methods predict inaccurate sizes or locations. Moreover, to more comprehensively demonstrate the effectiveness of the proposed method, we have also used a unique color for each bounding box in Figure 4 of the revised manuscript.
>
> Furthermore, we have experimented on the challenging open-vocabulary 3D instance segmentation, where the incorporation of the prior of surface closure substantially enhances the segmentation performance of the Gaussian-Grouping baseline [2] (see Table 6 in the revised manuscript). The rendering results of qualitative examples containing multiple object instances are shown in Figure 7 of the revised manuscript.  It can be seen that incorporating the prior of surface closure leads to fewer noise-like erroneous predictions.
>
> [1] Hu, Benran, et al. "Nerf-rpn: A general framework for object detection in nerfs." CVPR. 2023.
>
> [2] Ye, Mingqiao, et al. "Gaussian grouping: Segment and edit anything in 3d scenes." ECCV, 2024.

---

### Official Review · Reviewer_s6A7 · 2024-11-10

**Soundness:** 2
**Presentation:** 1
**Contribution:** 1
**Rating:** 3
**Confidence:** 5

**Summary:**

This paper proposed a 3D object detection method based on the Gaussian splatting. With the Gaussian splatting as input, the proposed deep model extracts the 3D object bounding boxes. Experiments show the proposed methods give higher recall and precision than the competing methods.

**Strengths:**

The proposed method takes advantage of the Gaussian splatting representation for 3D object detection. Gaussian splatting is a more compact representation than the point clouds.

**Weaknesses:**

- The paper combines several methods: (1) Official implementation of Gaussian splatting and (2) PointNet++. There is no new network structure
developed. The novelty is low.

- The overall network is not explicitly defined in the paper. please add this, otherwise it is impossible to check the details.

- The writing of the paper is quite unclear, especially the section 3.3. Theorem 1 is directly from the existing theorem. No need to include here. It is also confusing where this theorem is used anywhere in the paper. The equations are sloppy. Notations are a mess. The features
used two notations F and f. F^{cand} is not defined. f^{part} is also never defined anywhere in the paper. The equations from (6), (7), (11) need motivation and clarification.

- The experimental setting is not clearly defined. The authors used a dataset originally for 3D object detection in Nerf. It is not clear how the point cloud based methods, which need a point cloud input, are used in the experiment comparison.

- The qualitative examples of the proposed method show the quality of the 3D boxes is sometimes a lot worse than other competing methods. The boxes are often bigger and quite off. The competing methods also show many overlapping bounding boxes. No non-max suppression This may cause the lower numbers from the competing methods.

**Questions:**

See the questions in the weakness section.

---

> ### Author Response · Authors · 2024-11-22
> **Response (Part I)**
>
> We truly appreciate for your valuable comments. In the following, we provide responses to the concerns.
>
> **Q1: Novelty of comibining existing Gaussian Splatting and PointNet++.**
>
> A: Thank you for your insightful question. We would like to clarify that our core contribution aims at **tackling the drawback of simply combining Gaussian Splatting and PointNet++ (or other point-based architectures).**
>
> While our proposed method consumes the output of Gaussian Splatting and uses a PointNet++ backbone, such a simple combination followed by a detection head cannot overcome the numerous outlier Gaussians which can distract the point-based detectors, thus leading to sub-optimal results prefixed by ''G-'' in Table 1.
>
> To address the aforementioned challenge, we seek to leverage the surface-based representation of 3D Gaussians and design the Closure Inferring Module (CIM) which (1) probabilistically models the underdetermined Gaussians for partial surfaces and (2) coalesces them into a quantitative measurement of the quality of object proposals via calculating the overall surface closure, thereby providing an informative prior on refining the sub-optimal objectness deduction. This contributes to a substantially better detection accuracy than methods that simply combine 3D Gaussians with point-based detectors.
>
> Moreover, we have further validated the generality of such measurement on the task of challenging open-vocabulary 3D instance segmentation (shown in Table A). The incorporation of the prior on surface closure sizably enhances the segmentation of the object instances.
>
> ### Table A: Open-vocabulary 3D instance segmentation on the real-world LERF-Mask/figurines dataset
>
>
> | Methods                             | mIoU | mBIoU |
> |-------------------------------------|------|-------|
> | DEVA (ICCV'2023)           | 46.2 | 45.1  |
> | LERF (ICCV'2023)            | 33.5 | 30.6  |
> | SA3D (NeurIPS'2023)             | 24.9 | 23.8  |
> | LangSplat (CVPR'2024)        | 52.8 | 50.5  |
> | Gaussian-Grouping (ECCV'2024) | 69.7 | 67.9  |
> | **Gaussian-Grouping+Ours**             | **76.5** | **73.3**  |
>
> **Q2: Definition of the overall framework.**
>
> A: Thank you for your valuable suggestion. Apart from the overall framework in Figure 3 of our revised manuscript where we have supplemented more details, we have added more network details in Appendix B.1, including the backbone network, the querying-and-pooling function, the stacked MLP for proposal refinement.
>
> **Q3: Necessity of Theorem 1.**
>
> A: Thank you for your feedback on Theorem 1. Theorem 1 is aimed at establishing the theoretical foundation for our Closure Inferring Module (CIM). The flux value $\Phi$ in Theorem 1 quantifies how well these surfaces approximate a closed shape, thus informing the closure weight $S_k$ in Eqn. (10) which influences the holistic feature representation used for proposal refinement. In implementation, we use the quadrature form (Eqn. (9)) of Theorem 1 by computing the flux over the partial surfaces within each object proposal to assess the surface closure.
>
> We are sorry that the previous version of narration regarding Theorem 1 causes your concern. We have carefully re-organized the related elaboration which we hope will address your concern. Moreover, for the reader's convenience, we have moved the proof of Theorem 1 to the Appendix for brevity.
>
> **Q4: Definitions of $F^{cand}$ and $\mathbf{f}^{part}_k$.**
>
> A: In the original manuscript, $F^{cand}$ is initially defined as the feature of candidate Gaussians (see L207 in the original manuscript, L217 in the revised manuscript). $\mathbf{f}^{part}_k$ is defined as the partial-aware features of each proposal (see L211-L213 in the original manuscript, L220-L223 in the revised manuscript).

---

> ### Author Response · Authors · 2024-11-22
> **Response (Part II)**
>
> **Q5: Motivation and clarification of Eqn. (6)(7)(10) in the revised manuscript.**
>
> A: **For Eqn. (6)**, in the revised manuscript, we have noted that as stated in [1] inferring local surface features is underdetermined due to the supervision from 2D signals, leading to unstructured outliers with unreliable local surface descriptors. Therefore, we model such randomness in CIM by incorporating a variational term into the candidate features, establishing a probabilistic feature residual.
>
> **For Eqn. (7)**, we follow [2] to optimize the evidence lower bound (ELBO) of the variational auto-encoder. In the revised manuscript, we have included the reference to [2] for clarity.
>
> **For Eqn. (10)**, in the revised manuscript, we have noted that $S_k$ serves as the closure weight parameter which controls the support ($|\hat{\mathbf{\Phi}}_k|\downarrow, S_k\uparrow$) or suppression ($|\hat{\mathbf{\Phi}}_k|\uparrow, S_k\downarrow$) of $\mathbf{f}^{hol}_k$. Moreover, we have added illustration on such support or suppresion in Figure 3 of the revised manuscript.
>
>
> **Q6: How the point cloud based methods are used in the experiment.**
>
> A: For the point cloud based methods, the original 3D Gaussians $\mathbb{G}=${$\mathbf{g}_p=(\mathbf{\mu}_p,\mathbf{\Sigma}_p,\alpha_p, \mathbf{c}_p)$}$^P_p$ (generated by 3DGS+SuGAR) were used as the input, where the Gaussian center $\mathbf{\mu}_p$ was regarded as the input point-wise coordinate. $\mathbf{\Sigma}_p,\alpha_p, \mathbf{c}_p$ were concatenated and regarded as the input point-wise features. Except for the first layer whose input dimension was modified to fit the input Gaussians, we faithfully followed the network and training configurations of the point cloud based methods we compared with. In the revised manuscript, we have added the aforementioned details to Appendix B.1.
>
> **Q7: On qualitative examples.**
>
> A: **Quality of the 3D boxes**: Thank you for pointing this out. In the revised manuscript, we have removed the first row with lower quality in Figure 5, and used a unique color for each bounding box for intuitive presentation.
>
> **NMS**: We clarify that in the qualitative results, we have faithfully used the official implementation of FCAF3D and NeRF-RPN, both using NMS as post-processing. We also clarify that the usage of Non-Maximum Suppresion (NMS) only removes the boxes that are overlapped over a pre-set threshold. Therefore, the overlapping observed in the competing methods is due to this threshold, which may have caused your concern.
>
> [1] Antoine, Guédon, et al. "Sugar: Surface-aligned gaussian splatting for efficient 3d mesh reconstruction and high-quality mesh rendering." CVPR, 2024.
>
> [2] Kingma, Diederik P. "Auto-encoding variational bayes." ICLR, 2013.

---

### Author Response · Authors · 2024-12-02
**General Response**

We genuinely thank all the reviewers for their constructive feedbacks and suggestions, which have helped us improve the quality and clarity of our work. Based on the reviewers' comments, we have revised our manuscript as follows:

- Added experiments on 3D instance segmentation in Table 6, Figure 7 of Section 4 and Table 11 of Appendix E.2 based on the comments by reviewer **s6A7**, **XhiG**, **kAE9**.
- Improved the clarity of the motivative illustration in Figure 2 based on the comments by reviewer **nYGE**.
- Improved the clarity of the framework figure along with the caption in Figure 3 based on the comments by reviewer **s6A7**, **nYGE**, **kAE9**.
- Improved the clarity and coherence of the Closure Inference Module (CIM) in Section 3.3 based on the comments by reviewer **s6A7**,**kAE9**.
- Improved and enriched the qualitative illustrations in Figure 4, Figure 12 and Figure 13 based on the comments by reviewer **XhiG**, **nYGE**.
- Added ablational experiments on the quality of Gaussian representation in Table 5 of Section 4 and Table 9 of Appendix C.3 based on the comments by reviewer **XhiG**, **nYGE**.
- Added verification of the quality of the 3D-FRONT dataset in Appendix A.1 based on the comments by reviewer **kAE9**.
- Added technical details of Gaussian-Det network and point cloud based methods we compared with in Appendix B.1 based on the comments by reviewer **s6A7**.

We respond to each reviewer below to address the concerns. Please take a look and let us know if further clarification or discussion is needed. Also, we will ensure that all discussions, tables and illustrations in the current revision are included in future versions.

---

### Meta-Review · Area_Chair_zw22 · 2024-12-21

**Metareview:**

The proposed method, Gaussian-Det, uses Gaussian Splatting (3DGS) to represent the surfaces for multiview 3D object detection. Gaussian-Det formulates the input Gaussians as feature descriptors on partial surfaces. Outliers derived from 3DGS are reduced by the Closure Inferring Module, which estimates the probabilistic feature residuals for partial surfaces and combines them to produce 3D object proposals.

The paper received mixed reviews, with scores of 3, 5, 6, and 6. After the rebuttal, one reviewer increased the rating, resulting in final scores of 3, 6, 6, and 6. It should be noted that the reviewer who gave the score of 3 did not provide any further responses during the rebuttal and reviewer discussion phases even though the authors adequately addressed the reviewer. Additionally, some comments from this reviewer contained apparent errors, which suggests a potentially low-quality review. Consequently, the area chair decided to disregard this review when making the final decision. Given that the other three reviewers provided positive feedback and consider the paper to be above the acceptance threshold, the area chair recommends accepting the paper.

**Additional Comments On Reviewer Discussion:**

In addition to the improvements included in the revision, the authors are encouraged to present the experimental results of using 2DGS as the underlying representation for its better surface modeling quality.

---

### Decision · Program_Chairs · 2025-01-22

Accept (Poster)